# Pharmacological Interaction of Quercetin Derivatives of *Tilia americana* and Clinical Drugs in Experimental Fibromyalgia

**DOI:** 10.3390/metabo12100916

**Published:** 2022-09-28

**Authors:** Yara Elena Quinto-Ortiz, María Eva González-Trujano, Edith Sánchez-Jaramillo, Gabriel Fernando Moreno-Pérez, Salomón Jacinto-Gutiérrez, Francisco Pellicer, Alonso Fernández-Guasti, Alberto Hernandez-Leon

**Affiliations:** 1Laboratorio de Neurofarmacología de Productos Naturales, Dirección de Investigaciones en Neurociencias, Instituto Nacional de Psiquiatría Ramón de la Fuente Muñiz, Calz. México-Xochimilco 101, Col. San Lorenzo Huipulco, Tlalpan, Mexico City 14370, Mexico; 2Departamento de Farmacia, Facultad de Química, Universidad Nacional Autónoma de México, Ciudad Universitaria, Coyoacán, Mexico City 04510, Mexico; 3Laboratorio de Neuroendocrinología Molecular, Dirección de Investigaciones en Neurociencias, Instituto Nacional de Psiquiatría Ramón de la Fuente Muñiz, Calz. México-Xochimilco 101, Col. San Lorenzo Huipulco, Tlalpan, Mexico City 14370, Mexico; 4Laboratorio de Farmacología Conductual, Departamento de Farmacobiología, Centro de Investigación y de Estudios Avanzados (CINVESTAV) Sede Sur, Calz. de los Tenorios 235, Col. Granjas Coapa, Tlalpan, Mexico City 14330, Mexico

**Keywords:** fibromyalgia, antinociception, reserpine-induced myalgia, *Tilia americana* var. *mexicana*, flavonoids

## Abstract

Fibromyalgia (FM) is a pain syndrome characterized by chronic widespread pain and CNS comorbidities. *Tilia americana* var. *mexicana* is a medicinal species used to treat anxiety, insomnia, and acute or chronic pain. However, its spectrum of analgesic efficacy for dysfunctional pain is unknown. To investigate a possible therapeutic alternative for FM-type pain, an aqueous *Tilia* extract (TE) and its flavonoid fraction (FF) containing rutin and isoquercitrin were evaluated alone and/or combined with clinical drugs (tramadol—TRA and pramipexol—PRA) using the reserpine-induced FM model in rats. Chromatographic analysis allowed the characterization of flavonoids, while a histological analysis confirmed their presence in the brain. TE (10–100 mg/kg, i.p.) and FF (10–300 mg/kg, i.p.) produced significant and dose-dependent antihyperalgesic and antiallodynic effects equivalent to TRA (3–10 mg/kg, i.p.) or PRA (0.01–1 mg/kg, s.c.). Nevertheless, the combination of FF + TRA or FF + PRA resulted in an antagonistic interaction by possible competitive action on the serotonin transporter or µ-opioid and D_2_ receptors, respectively, according to the *in silico* analysis. Flavonoids were identified in cerebral regions because of their self-epifluorescence. In conclusion, *Tilia* possesses potential properties to relieve FM-type pain. However, the consumption of this plant or flavonoids such as quercetin derivatives in combination with analgesic drugs might reduce their individual benefits.

## 1. Introduction

Fibromyalgia (FM) is characterized by multifocal, widespread, chronic musculoskeletal pain [1]. It is a syndrome of unknown origin accompanied by symptoms such as sleep disorders, fatigue, and depression, among others [2]. Depending on the diagnostic criteria, the worldwide prevalence is 2–8% [3,4]. In women, it represents 70 to 90% of the established cases [5]. In 2010, the American College of Rheumatology approved a new and more comprehensive set of criteria, which includes both a Widespread Pain Index (WPI) that evaluates the number of reported painful body regions, and a Severity Scale (SS) that considers the most frequent concomitant symptoms. Clinical data developed in 2016 for diagnosis of FM describe the following criteria: (1) generalized pain in at least four out of five regions, (2) symptoms present for at least 3 months, (3) WPI ≥ 7 and SS ≥ 5 or WPI = 4–6 and SS ≥ 9. Additionally, the diagnosis of FM is valid in presence of other painful illnesses. These changes in diagnostic criteria decreased the difference in the prevalence ratio between women and men to 2:1 [6].

Regarding pharmacological therapy of FM, clinical drugs producing central inhibitory effects have been suggested as the most efficacious, such as those decreasing excitatory activity (neurotransmission of glutamate) or increasing inhibitory neurotransmission such as those of norepinephrine (NA), serotonin (5-HT), or gamma-aminobutyric acid (GABA) [7]. Individual administration of an analgesic drug acting at the central level, as well as the combination of more than one clinical analgesic drug or together with natural products, could be used as a strategy for FM-type pain relief [8,9,10,11,12]. This is because multiple therapeutic targets, as mechanisms of action, could be better than those of conventional individual pharmacotherapy known for the treatment of FM [13]. It is known that, in medicinal plants, several constituents are responsible for their pharmacological efficacy, producing potential interactions such as those displayed by flavonoids [12,14,15,16,17,18].

*Tilia americana* var. *mexicana* is an endemic Mexican species used in folk medicine due to its effects on the CNS. It is prepared as an infusion and used to treat anxiety, insomnia, headache, and to alleviate inflammation, arthritis, and other kinds of diseases associated with pain [16,19,20]. Preliminary studies reported the significant analgesic-like effects of different organic extracts of *Tilia* using a gouty arthritis model and the formalin test in rats [16,17]. Depressant effects of *Tilia* have been associated to the presence of flavonoid constituents [14,15,21] such as kaempferitrin, quercetin, isoquercetin, quercitrin, isoquercitrin, astragaline, rutin, and tiliroside, among the principal metabolite-producing CNS activities such as sedative, anxiolytic [15,21], antinociceptive, and anti-inflammatory pain [16].

Novel therapies are continuously developed by using validated animal models. The reserpine-induced experimental FM is a model considered to produce nociceptive responses such as hyperalgesia and allodynia, aside from sleeping disorders and depression comorbidities [22,23,24]. In this study, the potential effects of *Tilia* extract and a bioactive fraction containing flavonoids, administered alone and combined with tramadol (TRA) or pramipexol (PRA), were explored in the reserpine (RES)-induced FM-type pain in rats, corroborating its presence at the central level by epifluorescence histological examination and receptor interaction by *in silico* analysis.

## 2. Materials and Methods

### 2.1. Animals

Female and male Wistar rats (200–300 g body weight) bred in the vivarium of Instituto Nacional de Psiquiatría Ramón de la Fuente Muñiz were used in this study. Six rats were housed in each cage in a temperature- (22 ± 2 °C) and light-controlled room under a 12 h light/dark cycle (light on at 7:00 a.m.) with free access to tap water and Purina rat chow. All experimental procedures were performed per the Official Mexican Norm for care and handling animals (NOM-062-ZOO-1999), as well as in compliance with the NC17073.0 protocol approved by the Research Committee and Institutional Animal Ethical Committee (CONBIOETICA-09-CEI-010-20170316).

### 2.2. Plant Material

*Tilia americana* L. var. *mexicana* (Schltdl.) Hardin (Tiliaceae) inflorescences were collected in August 2017 in Morelia within the state of Michoacán, Mexico. Identification of the species was overseen by the botanist José Luis Contreras (0000-0002-7946-8858). A specimen number FCME131611 was deposited at the herbarium of Facultad de Ciencias, Universidad Nacional Autónoma de México, Mexico City, for future reference.

#### 2.2.1. Preparation of the Extract and the Bioactive Fraction

The air-dried and milled *Tilia* inflorescences (543.8 g) were initially macerated at room temperature (22 ± 2 °C) followed by the addition of hexane (250 mL), ethyl acetate (250 mL), and methanol (250 mL). Each time, the solvent was separated from the residues by gravity filtration and then vacuum-evaporated. Three successive extractions were carried out with each solvent to yield: hexane (0.28%), ethyl acetate (0.46%), and methanol (2.80%) extracts.

Eleven grams of the methanol extract were fractionated by open-column chromatography (OCC) with an increasing solvent gradient of polarity using hexane, ethyl acetate, and methanol. Solvents were vacuum-evaporated after fractionation. Twenty-one subfractions were obtained and pooled based on their spot similarities using thin-layer chromatography (TLC) [25] to obtain 9 pools. The presence of flavonoids was firstly analyzed by TLC using the reagent of natural products (NP/PEG) under UV light (365 nm) to corroborate the presence of quercetin derivatives (data showed in graphical abstract), after which, hydrolysis resulted in the presence of only the aglycone, as corroborated by TLC and high-performance liquid chromatography (HPLC).

#### 2.2.2. HPLC Analysis

Both the extract (TE, 10 mg/mL) and its flavonoid fraction (FF, 5 mg/mL) were analyzed by HPLC to determine their chromatographic profiles and characterize the most abundant flavonoids. The HPLC analysis was carried out using Waters Acquity UPLC H-Class liquid chromatography equipment fitted with a Waters photodiode array detector (UPLC, Acquity Waters, Singapore) and Empower chromatographic software version 3 (Waters, Milford, MA, USA). Separation was performed with a Symmetry C-18 column (100 Å, 150 mm × 4.6 mm, 5 mm, Waters, Wexford, Ireland) with the thermostat at 43 °C. The mobile phase was a gradient system consisting of Milli-Q water acidified with 0.1% phosphoric acid (solvent A) and HPLC-grade methanol (solvent B). The initial gradient mixture was 80% A:20% B during a total time of 12 min and a constant flow rate of 1.0 mL/min; solvent B’s concentration was gradually increased to 100% and, finally, was returned to the initial concentrations (80%:20%). The detection wavelength used was 254 nm. Commercial flavonoids (rutin, isoquercitrin, and quercetin, 1 mg/mL) were used as a reference to compare their corresponding retention times and maximum absorptions. All samples were dissolved in methanol (HPLC-grade) and filtered through 0.22 μm filters (GHP, Acrodisc 13, Waters) before being injected into the chromatograph.

#### 2.2.3. Presence of Flavonoid Derivatives of Quercetin by Brain Histological Analysis

The presence of flavonoids, such as quercetin aglycone and derivates, was corroborated by their natural epifluorescence observed in brain areas after histological analysis. Rats receiving quercetin or rutin at 30 and 100 mg/kg, i.p., were euthanized and their brains were quickly dissected on dry ice and stored at −80 °C until histological analysis. Rostro–caudal coronal sections from bregma −0.60 mm to −2.92 mm [26] were cut at a 20 µm of thickness on a cryostat (Leica Biosystems Nussloch GmbH CM3050S, Nussloch, Germany), adhered to Superfrost/Plus slides (Fisher Scientific, Waltham, MA, USA), each 120 µm apart and stored at −80 °C until used. Thawed sections were observed with epifluorescence using a Leica DM 1000 LED fluorescence microscope with a HI PLAN 10X/0.25 or 40X/0.65 objective and bandpass filter sets of 340–380/dichromatic mirror 400/barrier LP 425 for DAPI emission (Leica Microsystems AG, Wetzlar, Germany). The images were captured using a digital FireWire camera (DFC450C, Leica Biosystems Nussloch GmbH, Nussloch, Germany) and Leica Application Suite software. Sections were exposed to DAPI blue filter sets, and the resulting images were processed in Adobe Photoshop CS6 software (Adobe Inc., San Jose, CA, USA) using an iMac computer.

### 2.3. Drugs and Reagents

Reserpine (RES), Tween 80, PRA hydrochloride hydrate, acetic acid, and flavonoid standards (STDs: rutin, isoquercitrin, and quercetin, 95% purity) were purchased from Sigma-Aldrich (St. Louis, MO, USA). RES was dissolved in acetic acid and diluted to a final concentration of 0.5% with distilled water and administered subcutaneously (s.c.). TRA (Antibióticos de México, S.A. de C.V. AMSA, Mexico City, Mexico), as a clinical drug reference, TE, FF, and individual flavonoids were resuspended in 0.5% Tween 80 in distilled water and administered intraperitoneally (i.p.). PRA (s.c.) was diluted in distilled water and used as reference drug too. Drugs were freshly prepared on the day of the experiments. All treatments were administered to rats in a volume of 1 mL/kg.

### 2.4. Reserpine (RES)-Induced FM Model

RES-induced myalgia was carried out as previously reported [22,23]. Hernandez-Leon et al. [23] described that RES administration equally affects female and male rats, so for this study, female rats were arbitrarily selected in all phases of the estrous cycle. Animals were allowed to habituate to the experimental conditions of the hyperalgesia and allodynia measurements for three days before the induction of the FM-type condition. RES (1 mg/kg) was s.c. administered into the loose skin over the neck every 24 h over three consecutive days. Then, measurements of thresholds for muscle pressure and response to tactile and cold stimuli were registered at baseline (t = 0) and 30, 60, 120, 180, and 240 min after the administration of the vehicle, reference drug, extract, fractions, individual flavonoids, or individual flavonoids in combination with the reference drugs (Figure 1, timeline).

#### 2.4.1. Experimental Groups

Animals were individually housed in acrylic cages (17 cm width × 27 cm length × 20 cm height). Muscle hyperalgesia or tactile and cold allodynia were measured in rats receiving RES (FM group) or vehicle (control group). To determine antinociceptive activity, different groups of rats received doses following a logarithmic increase, specifically: TE (10, 30, and 100 mg/kg, i.p.), FF (10, 30, 100, and 300 mg/kg, i.p), TRA (3 and 10 mg/kg, i.p.), or PRA (0.01, 0.1, 0.5, and 1.0 mg/kg, s.c.). Quercetin (562 mg/kg, i.p.) was used as a reference for flavonoids. To analyze pharmacological interaction, FF was co-administered with each clinical drug using the minimal individual effective dose. Combinations consisted of FF (10 mg/kg, i.p.) + TRA (3 mg/kg, i.p.) or FF (10 mg/kg, i.p.) + PRA (0.1 mg/kg, s.c.) (Figure 1, timeline).

#### 2.4.2. Threshold for Muscle Pressure (Mechanical Hyperalgesia)

Mechanical hyperalgesia (tenderness to palpation) is frequently used to estimate muscular pain thresholds in patients with FM. In the experimental model, the threshold for muscle pressure was measured in rats using the Randall–Selitto apparatus (Ugo-Basile, Varese, Italy). To avoid involuntary movements, rats were immobilized inside a sock, and then the right hind limb was positioned so that linear pressure could be applied to the mid-gastrocnemius muscle. The behavioral response was considered until the limb was withdrawn or a vocalization occurred. The maximum force applied was set to 250 g. Three consecutive trials were performed and the threshold for muscle pressure was considered as the average of them. An inter-stimulus/inter-trial interval of at least 1 min was considered to avoid sensitization of the hind limb to mechanical stimulation [22,23].

#### 2.4.3. Threshold for Tactile Response (Tactile Allodynia)

Tactile allodynia is also a characteristic response of FM disease. The threshold for the tactile response was determined using von Frey filaments (North Coast Medical, Morgan Hill, CA, USA) following the up-and-down method [27]. Rodents were individually placed inside an acrylic cylinder on an elevated metal grid. The filaments (2, 4, 6, 8, 10, and 15 g) were applied underneath the grid into the plantar surface of the right hind paw. Because of filament pressure, strong paw withdrawal, licking, and paw shaking were registered as positive responses. On the other hand, the lack of paw withdrawal within 6 s was a negative response. After the 2 g filament application, if a positive response occurred, the next smaller filament was applied; when a negative response occurred, the next higher filament was applied. The test continued until four responses were completed after the first observed change in the behavioral response. The response pattern obtained was converted to the threshold for the tactile response according to the following formula: 10[Xf − kδ]/10,000, where Xf is the log value of the von Frey filament used, k is a tabular value for the response pattern, and δ equals the mean difference between stimuli (log units) [23].

#### 2.4.4. Threshold for the Cold Stimulus (Cold Allodynia)

Studies in FM patients have confirmed an increased sensitivity to cold and warm sensations [28]. The threshold for the cold stimulus response was determined using the acetone spray test, a modified method from Choi et al. [29]. Rats were individually placed inside an acrylic cylinder on an elevated metal grid. Using an insulin syringe with a drip adapter, 50 μL of acetone was applied onto the middle of the dorsal surface of the right hind limb. The response time of the right hind paw to licking, shaking, flinching, or lifting was counted over a 60 s period following the spray. The threshold for the cold stimulus response was considered as the average of three measurements with an inter-stimulus interval of at least 5 min [30].

### 2.5. In Silico Analysis

To support a possible direct interaction of quercetin on the receptors of TRA and PRA such as the µ-opioid receptor, serotonin transporter, and D_2_ dopamine receptor, respectively, the crystal structure of the compounds quercetin, morphine, and PRA was obtained from the PubChem database. These structures were protonated using the Avogadro software V1.2.0 (Avogadro Chemistry, Pittsburgh, PA, USA) for a pH of 7.4 and, subsequently, the minimum-energy spatial configuration was determined using the Merck Molecular Force Field (MMFF94, Merck Research Laboratories, Boston, MA, USA). The protein structure of the µ-opioid receptor (4DRL), serotonin transporter (6VRH), or D_2_ (6CM4) dopamine receptor was obtained from the Protein Data Bank (PDB, https://www.rcsb.org/ accessed on 28 July 2022); for each case, a resolution of ≤3.3 Å was selected. Then, docking was performed by the CB-Dock tool [31]; results from CB-Dock tool software V2.0 (Structural Bioinformatics Research group, Chengdu, China) were contrasted with UCSF Chimera 1.16 (Resource for Biocomputing, Visualization, and Informatics University of California, San Francisco, CA, USA) for protein preparation and AutoDock Vina 1.1.2 (Oleg Trott, La Jolla, CA, USA).

### 2.6. Statistical Analysis

Results are expressed as the mean ± standard error of the mean (SEM) of at least 6 repetitions. Data from the time-course curves (TCC) were analyzed by a two-way analysis of variance (ANOVA) of repeated measures (factors: time and treatment). The area under the curve (AUC) was obtained from the respective TCC of the nociceptive thresholds by the trapezoidal method and analyzed by a one-way ANOVA (factor: treatment). In both ANOVAs, the post-hoc Tukey’s test was applied. The F-value requires two parameters: degrees of freedom of numerator (number of treatments minus one) and degrees of freedom denominator (number of experimental subjects minus number of treatments), where the intersection of numerator and denominator contains the critical F-value. The F-value obtained in the corresponding one-way or two-way ANOVA analysis and the degrees of freedom (df) were required to be contrasted with the F-table critical values for a significance level of 0.05. *p* < 0.05 was considered significant using the GraphPad Prism version 8.0 software (GraphPad Software Inc., La Jolla, CA, USA).

## 3. Results

### 3.1. Phytochemical Analysis

#### 3.1.1. Flavonoid Characterization

According to thin-layer chromatography analysis, using quercetin as a reference standard, both *Tilia* extract (TE) and flavonoid fraction (FF) samples demonstrated the presence of abundant flavonoid constituents, as previously reported in this species [14,21] (Figure 2A). The presence of flavonoids rutin and isoquercitrin was characterized in a fraction of the *Tilia* identified by HPLC analysis of FF (10 mg/mL) (Figure 2B) and corroborated as glycosides of quercetin after an acid hydrolysis (Figure 2C).

#### 3.1.2. Epifluorescence of Flavonoids

In order to corroborate the presence of flavonoids into the brain, rats receiving 30 or 100 mg/kg, i.p., of the quercetin (aglycone) or its glycoside rutin were euthanized at 30 min, 4 h, and 24 h after treatments. Then, cerebral coronal sections were obtained and observed with a microscope to observe the presence of self-fluorescence of quercetin and rutin, such as red particles around the third ventricle, by using a blue emission filter from the first 30 min after treatment. An accumulation of fluorescent particles was also observed in the hypothalamic arcuate nuclei in samples of 100 mg/kg, i.p., at 4 h. Self-fluorescence remained longer in the brain, since it was also noticed in sections from rats euthanized 24 h after treatment (Figure 3).

### 3.2. Pharmacological Evaluation

#### 3.2.1. Antinociceptive Effects of *Tilia* Extract (TE) and Flavonoid Fraction of *Tilia* (FF)

Reserpine injection in rats previously administered with vehicle produced the expected nociceptive response in muscle hyperalgesia, as well as in tactile and cold allodynia (Figure 4A–C, respectively). Hyperalgesia was significantly reduced with all doses of TE (10–100 mg/kg), as observed in time-course curves (TCC) from 30 to 240 min after i.p. injection (Figure 4A), showing a recovery from 38% to 83% (F_4,33_ = 81.10, *p* < 0.0001) (Figure 4D). A similar antinociceptive response was obtained in the tactile allodynia at 30 (62%) to 100 mg/kg (68%) (F_4,33_ = 59.60, *p* < 0.0001) (Figure 4B,E). While in cold allodynia, all doses of TE produced a total inhibition (100%) of nociception (Figure 4C,F) (F_4,33_ = 40.05, *p* < 0.0001).

A dose–response curve of FF (10 to 300 mg/kg) was carried out in comparison to the standard quercetin (562 mg/kg). FF (30 to 300 mg/kg) produced significant antihyperalgesic effects after the first 30 min of its i.p. injection (Figure 5A–C) and ranged in its response from 36% to 48% (Figure 5D) (F_6,37_ = 14.18, *p* < 0.0001), whereas antiallodynic effects were observed, using mechanical stimulus, from 34% to 57% (Figure 5E) (F_6,37_ = 11.99, *p* < 0.0001) and 85% to 92% in thermal stimulus (Figure 5F) in comparison to those of quercetin, which produced an 81% antihyperalgesic effect and a 69% mechanical (Figure 5D,E) or 92% thermal (Figure 5F) antiallodynic response (F_6,37_ = 53.20, *p* < 0.0001).

#### 3.2.2. Antihyperalgesic and Antiallodynic Effects of Tramadol (TRA) and Pramipexole (PRA)

TRA (10 mg/kg, i.p.) produced a significant antihyperalgesic effect from the first 30 to 60 min after injection (Figure 6A). The AUC showed a 29% relief at 240 min (Figure 6D) (F_3,28_ = 56.55, *p* < 0.0001). In tactile allodynia, the same dosage of TRA produced a significant effect from 30 to 120 min (Figure 6B), remaining in 54% of the maximal response (Figure 6E) (F_3,28_ = 26.10, *p* < 0.0001). In the case of cold allodynia, TRA (3 and 10 mg/kg) produced a 58% to 89% antiallodynic effect (F_3,28_ = 42.25, *p* < 0.0001) (Figure 6F).

Regarding PRA (1 mg/kg), a maximal antihyperalgesic effect of 39% (F_5,40_ = 21.50, *p* < 0.0001) was produced at 30 min post-injection, which exponentially decreased to reach basal values (Figure 7A,D). In the tactile allodynia, PRA (1 mg/kg) produced a significant effect of 87% (F_5,40_ = 40.14, *p* < 0.0001) (Figure 7B,E). Finally, in cold allodynia, PRA produced an antinociceptive effect from 64% to 95% (F_5,40_ = 53.56, *p* < 0.0001) at all tested doses (Figure 7C,F). In both tactile and cold allodynia, the effect was observed from the first 30 min and remained until the end of the experiment (240 min after s.c. injection).

#### 3.2.3. Combination of FF with TRA and PRA

To explore pharmacological interactions, the sub-effective doses of FF (10 mg/kg) and TRA (3 mg/kg) or PRA (0.1 mg/kg) were combined and evaluated in the FM rats. Administration of combinations of FF + TRA or FF + PRA did not modify the muscle hyperalgesia threshold (Figure 8A,D) or tactile allodynia (Figure 8B,E). Regarding cold allodynia, both combinations produced a significant decrease compared to the response obtained in the individual administration (Figure 8C,F).

#### 3.2.4. *In Silico* Analysis of Quercetin, TRA, and PRA on the Serotonin Transporter and µ-opioid and D_2_ Receptors

An *in silico* assessment using a molecular docking approach for target validation was conducted to contribute to a theoretical explanation of the drug–receptor relationship obtained in the *in vivo* protocol. The results showed that quercetin possesses a binding energy in the range of −9.4 to −7.2 kcal/mol for the µ-opioid receptor (Figure 9A), −8.9 to −7.6 kcal/mol for the serotonin transporter (Figure 9C), and −9.5 to −7.5 kcal/mol for the D_2_ dopamine receptor (Figure 9E). In comparison, TRA showed −6.8 to −5.8 kcal/mol for the µ-opioid receptor (Figure 8B) and −6.3 to −5.6 kcal/mol for the serotonin transporter (Figure 9D), and PRA showed −7 to −5.2 kcal/mol for the D_2_ dopamine receptor (Figure 9F). All the results together suggest better affinity of quercetin versus TRA or PRA for resulting in an antagonistic interaction as corroborated in the *in vivo* antihyperalgesic and antiallodynic responses (Figure 8A–F).

## 4. Discussion

Fibromyalgia (FM) is a chronic disease of unknown etiology, whose manifestation is widespread musculoskeletal pain and some persistent comorbid symptoms which decrease the quality of life in patients and represent a significant socio-economic impact on the population [32]. A multidisciplinary therapeutic approach, which includes drugs and alternative therapies such as light, thermal and electrical stimulation, natural products, and body exercise, is considered as a possible strategy [11,33].

The evidence for the effectiveness of current therapy remains unsatisfactory and insufficient due to the plurality of perceptions related to FM within the medical practice as well as the limited animal models that mimic this kind of pain and their main comorbidities [34]. An imbalance in the content of biogenic amines such as serotonin, noradrenaline, and dopamine are known to play an important role in the pathogenesis of pain at the CNS level in FM [35,36,37]. Because of this, the experimental model used in this investigation considers the depletion of biogenic amines induced by RES to correlate with the onset muscle hyperalgesia and allodynia by peripheral and spinal nociceptive mechanisms, which has been behaviorally, pharmacologically, and neurochemically characterized [22,23,38,39,40]. It is known that this experimental model also mimics clinical characteristics such as the appearance of sleep disorders [24,41] and depressive-like behaviors [22,42].

TRA or PRA as monotherapy and/or in combination are clinical analgesic drugs used in FM therapy resulting in moderate efficiency [10]. Although it is well-known that TRA produces an analgesic effect by an agonistic action on the μ-opioid receptor, it is also considered an atypical action as a reuptake inhibitor of serotonin and noradrenaline. PRA is an agonist acting on the D_2_, D_3_, and D_4_ dopamine receptors. Both clinical drugs have been evaluated and compared with placebo in trials of patients with severe FM, where TRA has shown moderate analgesia in combination with paracetamol [43,44], while PRA has been explored in patients concomitantly medicated [45,46].

An amount of 40 to 60 % of FM patients do not respond well to drug therapy. For this reason, the search for new compounds derived from natural products capable of inhibiting pain, such as medicinal plants or their secondary metabolites, has been promoted worldwide [47]. Nevertheless, evidence for the safety and effective use of various plant extracts remains scarce [11]. Preclinical studies suggest the potential use of *Tilia americana* for the FM therapy due to its properties on the CNS as an antinociceptive, anticonvulsant, and its anxiolytic/sedative-like effects [16,21,48]. In addition, some phytochemical studies reported that anxiolytic/sedative-like effects are due to the presence of compounds, mainly from flavonoids and as derivatives of quercetin and kaempferol [14,15].

In this study, the chromatographic profiles and epifluorescence histological analysis of TE and FF indicated the presence of glycosylated derivatives of quercetin such as rutin and/or isoquercitrin that, after an acid hydrolysis, result in the aglycone quercetin. This agrees with previous studies that stated that flavonoids are the main components in the polar extracts from the *Tilia* genus [15,21,48]. Not only rutin and isoquercitrin, but also kaempheritrin, quercitrin, astragalin, and tiliroside have been corroborated as major components in several *Tilia* species such as *T. cordata*, *T. rubra*, *T. argentea*, and *T. platyphyllos* [15,16,49]. Although the presence of these similar constituents in great abundance was confirmed in this study, we do not discard the notion that other flavonoids or constituents in lower concentrations might also participate in the antinociceptive effect of the extract.

Our results from the pharmacological assessment provide evidence of the potential of this species in reducing hyperalgesia and allodynia in experimental FM in rats. Preliminary studies reported antinociceptive and anti-arthritic effects of TE [16,17] in part due to a synergistic effect of its flavonoid compounds [48,50]. However, regarding the individual flavonoids, antinociceptive effects were reported for rutin in the formalin, glutamate, and hot-plate tests [50,51,52,53], and beneficial effects were observed in animal models of inflammation-like arthritis [54] and peripheral neuropathy [55]. In the case of quercitrin, it has been reported as an anti-inflammatory in a rat colitis model [56], in acute lung injury [57], and in visceral pain [58]. As for isoquercitrin, anti-inflammatory effects were produced in LPS-induced inflammation [59] and allergic asthma [60]. Antinociceptive effects of FF in our present investigation suggest a greater synergistic interaction between the identified, and possibly other non-identified compounds, that improves the antinociceptive effect observed, in comparison to TE.

The efficacy of PRA on pain, depression, and overall health status in patients with FM [45,61] in our results resembled those reported by Nagakura et al. [22] concerning the antihyperalgesic and antiallodynic efficiency of doses of 0.1 and 1 mg/kg in the RES-induced nociceptive thresholds. To the best of our knowledge, the effect of PRA in cold allodynia has not been previously reported in this FM model. Our results demonstrate, for the first time, a sustained and dose-dependent effectiveness in this threshold, modified in the presence of TE and FF, too. TRA antinociceptive activity involves noradrenergic and serotonergic neurotransmission by using the tail-flick and hot-plate tests [62]. Kaneko et al. [63] described its antiallodynic effect in the RES-induced FM model in rats. Our results are consistent with this report where the TRA (10 mg/kg) effect was maintained for the first two hours and 30 min after a dosage of 3 mg/kg. Furthermore, our results demonstrated that TRA (3 mg/kg) elicits significant antihyperalgesic effects and antiallodynic activity in cold stimulus.

Drug interaction is a common clinical practice, since combined therapies can generate beneficial efficiency and/or reduce undesirable effects by administering together lower doses of individual natural products and/or drugs to produce a similar or major pharmacological response [64,65,66]. Pain relievers might produce synergy, which is defined as a better response when two or more agents work together in combination [66,67]. Additive synergism is the theoretically expected effect of the combination of multiple drugs, whereas potentiation synergy occurs when the effect is greater than the foreseen additive effect of the individual drugs [67]. However, when the combined effect of compounds is lesser than the expected one, an antagonism occurs, and it is often considered a non-beneficial drug interaction [68].

Our present study contributes to the development of preclinical scientific evidence for proposal and/or caution when combinations of medicinal plants prepared in infusions/tinctures such as *Tilia americana* are co-administered with clinical drugs for pain treatment in FM. Our results showed that a significant antinociceptive flavonoid (quercetin derivatives)-rich fraction of this species did not produce a positive drug interaction when combined with TRA or PRA in the RES-induced FM, neither for hyperalgesia nor for allodynia, in part because to their CNS influence. It is known that at least one of the mechanisms of action of flavonoids found in FF is similar to that of TRA or PRA, resulting in an antagonistic result when combined. Both rutin and TRA can influence an inhibition in serotonin and norepinephrine reuptake [69,70]. In a similar manner, activation of the opioidergic system has been reported for rutin, quercetin, and TRA [52], whereas quercetin and PRA can act as D_2_ and D_3_ dopamine receptor agonists [71]. The *in silico* analysis in this investigation emphasized the possibility of a direct interaction of quercetin and derivatives on µ-opioid and D_2_ receptors and the serotonin transporter to produce an antagonistic response. Regarding isoquercitrin, its mechanism of action is not well-described, but it can downregulate opioid receptors [72], avoiding the inhibitory action of this neurotransmission. A limitation of this study is that only a few doses were explored. An isobolographic and surface of interaction analysis could clarify if other proportions are able to produce different pharmacological interactions.

To support that the interaction of TRA on dopamine receptors and PRA on µ-opioid receptors or the serotonin transporter do not exist as possible molecular targets, a docking analysis was included in the Appendix A.

## 5. Conclusions

In conclusion, central nervous system effects of *Tilia americana* var. *mexicana* are reinforced by the present results, since significant antihyperalgesic and antiallodynic responses were demonstrated in an experimental FM model. The presence of quercetin-derivative flavonoids, as part of the responsible constituents, was confirmed in the brain tissue by epifluorescence, corroborating their importance and potential bioactivity and supporting their influence as a useful alternative for therapeutic treatment of FM-type pain. However, their administration together with clinical analgesic drugs such as tramadol or pramipexol, both used for their moderate efficacy against this syndrome, is not recommended, since they could reduce the benefits obtained in their individual administration.

## Figures and Tables

**Figure 1 metabolites-12-00916-f001:**
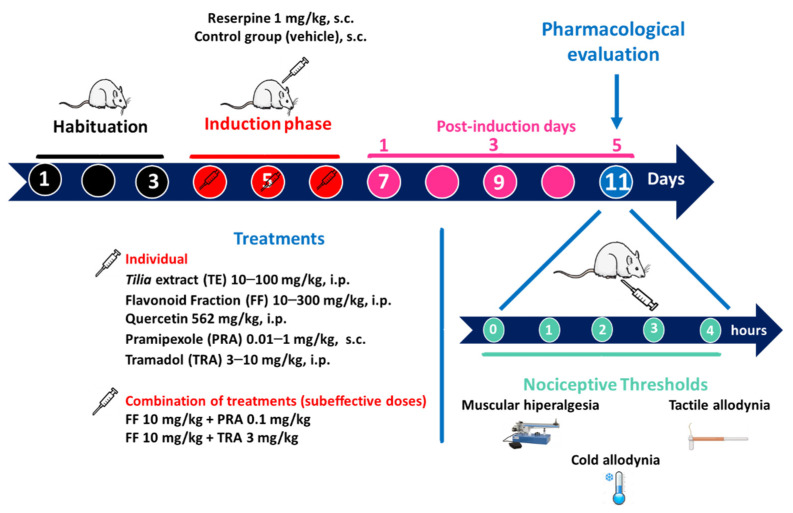
Timeline of the experimental design of *Tilia* fraction administration alone and in combination with pramipexole (PRA) and tramadol (TRA).

**Figure 2 metabolites-12-00916-f002:**
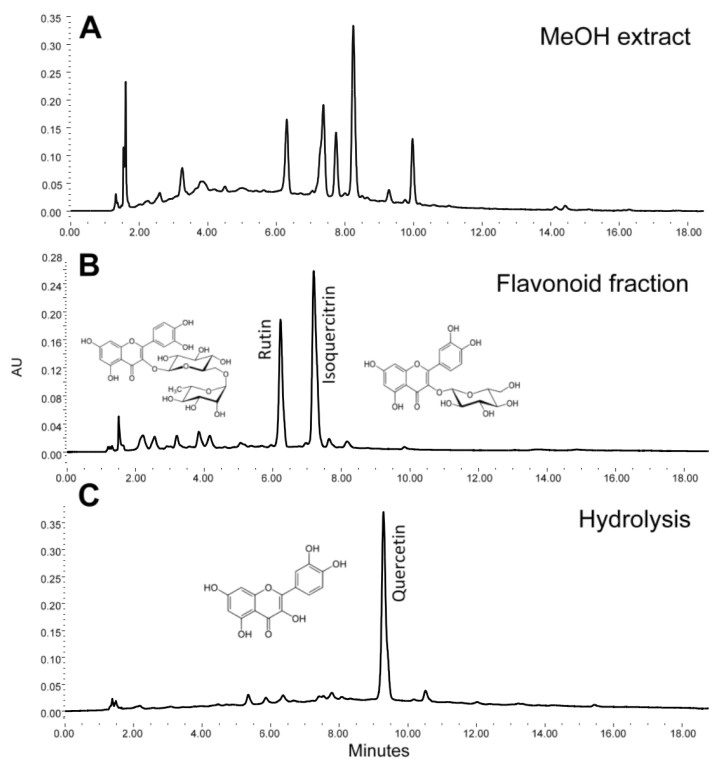
Chromatographic HPLC profile of the *Tilia* MeOH extract (**A**), flavonoid fraction (FF) containing glycosides rutin and isoquercitrin (**B**), and the aglycone quercetin as a result of an acid hydrolysis of the fraction (**C**).

**Figure 3 metabolites-12-00916-f003:**
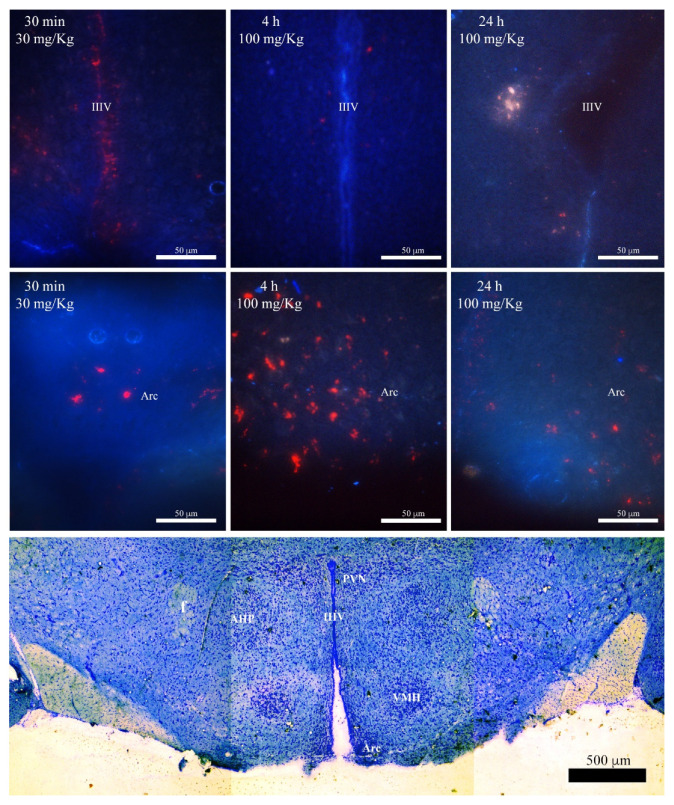
Microphotographs of brain from adult rats receiving 30 or 100 mg/Kg, i.p., of the flavonoid quercetin (aglycone, upper panel) and its glycoside rutin (intermediate panel). Representative coronal sections at 20 µm, around bregma −2.16 mm (bottom panel), were stored at −80 °C and then thawed to be observed with a blue filter for DAPI (UV excitation 340–380 nm/emission 425 nm) under the epifluorescence microscope. Horizontal micrograph at the bottom shows the location of the hypothalamic nuclei where quercetin and rutin were observed (Arc = arcuate nucleus) and the third ventricle (IIIV) in three contiguous images of a brain section stained with Nissl. Other hypothalamic nuclei negative to quercetin and rutin are also shown. AHP = anterior hypothalamic area; PVN = paraventricular nucleus; WMH = ventromedial hypothalamic nuclei; f = fornix. Scale bars in upper panels = 50 µm; scale bar at the bottom = 500 µm.

**Figure 4 metabolites-12-00916-f004:**
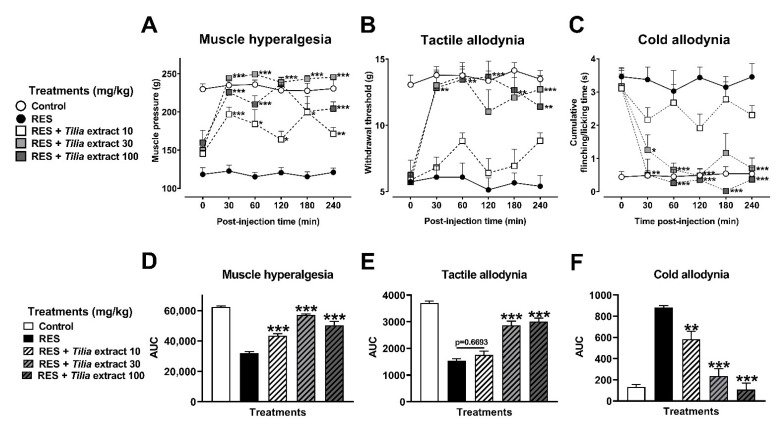
Antihyperalgesic and antiallodynic effect of the *Tilia* extract on the reserpine (RES)-induced nociceptive threshold. Time-course curves of the threshold for muscle pressure, shown as the force in grams supported in the right gastrocnemius muscle (**A**), the tactile response, shown as the force in grams supported on the plantar surface of the right hind limb (**B**), and cold allodynia, shown as the cumulative response time to cold stimulus (acetone) on the dorsal surface of the right hind limb for 1 min (**C**). Area under the curves of the muscle pressure threshold (**D**), tactile response (**E**), and cold allodynia (**F**). One-way (**D**–**F**) or two-way (**A**–**C**) ANOVA followed by Tukey´s test, * *p* < 0.05, ** *p* < 0.01, *** *p* < 0.001 vs. RES group.

**Figure 5 metabolites-12-00916-f005:**
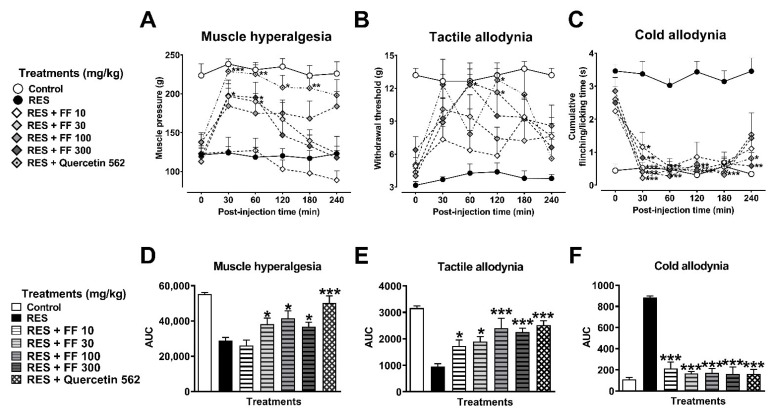
Antihyperalgesic and antiallodynic effect of a flavonoid fraction of *Tilia* (rutin and isoquercitrin) on the reserpine (RES)-induced nociceptive threshold. Time-course curves of the threshold for muscle pressure, shown as the force in grams supported in the right gastrocnemius muscle (**A**), the tactile response, shown as the force in grams supported on the plantar surface of the right hind limb (**B**), and cold allodynia, shown as the cumulative response time to cold stimulus (acetone) on the dorsal surface of the right hind limb for 1 min (**C**). Area under the curves of the muscle pressure threshold (**D**), tactile response (**E**), and cold allodynia (**F**). One-way (**D**–**F**) or two-way (**A**–**C**) ANOVA followed by Tukey´s test, * *p* < 0.05, ** *p* < 0.01, *** *p* < 0.001 vs. RES group.

**Figure 6 metabolites-12-00916-f006:**
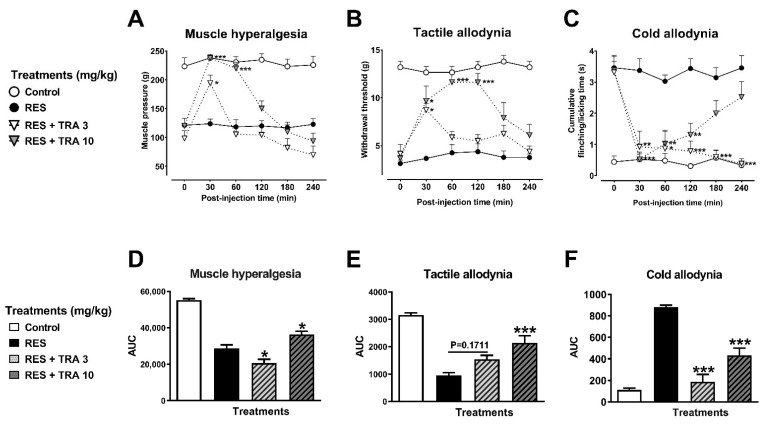
Antihyperalgesic and antiallodynic effect of tramadol (TRA) on the reserpine (RES)-induced nociceptive threshold. Time-course curves of the threshold to muscle pressure, shown as the force in grams supported in the right gastrocnemius muscle (**A**), the tactile response, shown as the force in grams supported on the plantar surface of the right hind limb (**B**), and cold allodynia. shown as the cumulative response time to cold stimulus (acetone) on the dorsal surface of the right hind limb for 1 min (**C**). Area under the curves of the muscle pressure threshold (**D**), tactile response (**E**), and cold allodynia (**F**). One-way (**D**–**F**) or two-way (**A**–**C**) ANOVA followed by Tukey´s test, * *p* < 0.05, ** *p* < 0.01, *** *p* < 0.001 vs. RES group.

**Figure 7 metabolites-12-00916-f007:**
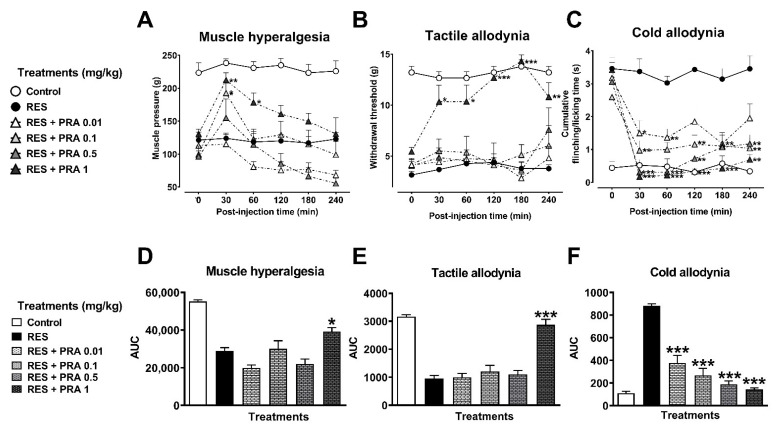
Antihyperalgesic and antiallodynic effect of pramipexole (PRA) on the reserpine (RES)-induced nociceptive threshold. Time-course curves of the threshold to muscle pressure, shown as the force in grams supported in the right gastrocnemius muscle (**A**), the tactile response, shown as the force in grams supported on the plantar surface of the right hind limb (**B**), and cold allodynia, shown as the cumulative response time to cold stimulus (acetone) on the dorsal surface of the right hind limb for 1 min (**C**). Area under the curves of the muscle pressure threshold (**D**), tactile response (**E**), and cold allodynia (**F**). One-way (**D**–**F**) or two-way (**A**–**C**) ANOVA followed by Tukey’s test, * *p* < 0.05, ** *p* < 0.01, *** *p* < 0.001 vs. RES group.

**Figure 8 metabolites-12-00916-f008:**
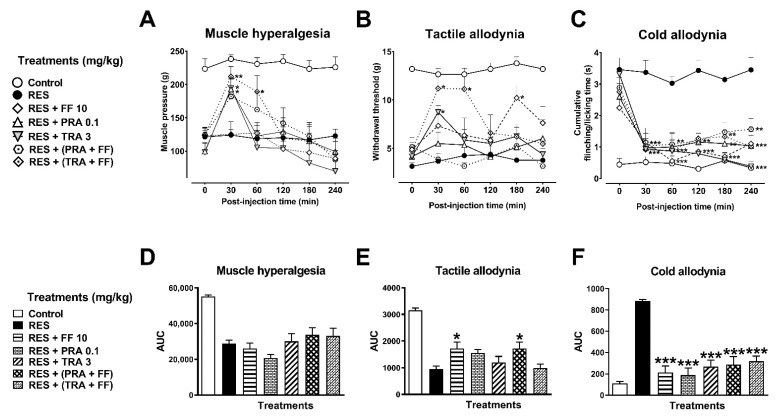
Antihyperalgesic and antiallodynic effects of the flavonoid fraction of *Tilia* (FF) in combination with pramipexole (PRA) or tramadol (TRA) on the reserpine (RES)-induced nociceptive threshold. Time-course curves of the threshold to muscle pressure, shown as the force in grams supported in the right gastrocnemius muscle (**A**), the tactile response, shown as the force in grams supported on the plantar surface of the right hind limb (**B**), and cold allodynia, shown as the cumulative response time to cold stimulus (acetone) on the dorsal surface of the right hind limb for 1 min (**C**). Area under the curves of the muscle pressure threshold (**D**), tactile response (**E**), and cold allodynia (**F**). One-way or two-way ANOVA followed by Tukey´s test, * *p* < 0.05, ** *p* < 0.01, *** *p* < 0.001 vs. RES group.

**Figure 9 metabolites-12-00916-f009:**
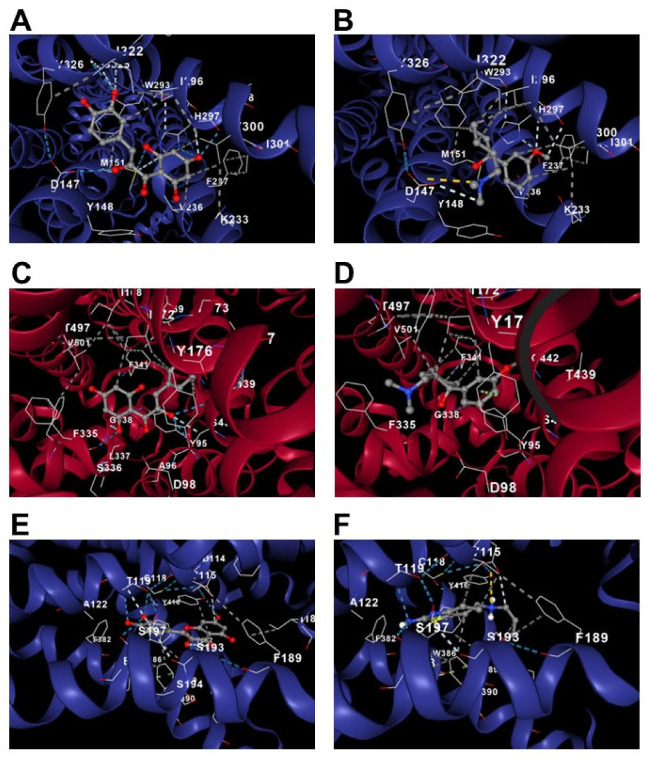
*In silico* evaluation of the interaction of quercetin with the µ-opioid receptor (**A**), serotonin transporter (**C**), and D_2_ dopamine receptor (**E**) in comparison with the interactions of the tramadol with µ-opioid receptor (**B**), serotonin transporter (**D**), and pramipexole with D_2_ dopamine receptor (**F**). It is observed that quercetin–µ-opioid complex shared 13 steric interactions with tramadol–µ-opioid complex (ASP114, VAL115, CYS118, THR119, ALA122, PHE189, SER193, SER197, PHE382, TRP386, PHE389, PHE390, HIS393, and TYR416) (**A**,**B**). The quercetin–serotonin transporter complex shared 5 steric interactions with the tramadol–serotonin transporter complex (ILE172, TYR176, PHE341, SER438 and THR439) (**C**,**D**). Finally, the quercetin–D_2_ dopamine complex shared 14 steric interactions with pramipexole–D_2_ dopamine complex (ASP114, VAL115, CYS118, THR119, ALA122, PHE189, SER193, SER197, PHE382, TRP386, PHE389, PHE390, HIS393, and TYR416).

## Data Availability

All data generated or analyzed during this study are included in this published article (and its Appendix A).

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
