# Peer review of "Pharmacological Interaction of Quercetin Derivatives of Tilia americana and Clinical Drugs in Experimental Fibromyalgia"

_metabolites, 2022, doi:10.3390/metabo12100916_

Round 1

Reviewer 1 Report

The manuscript has been written well, authors must mention the full details of the docking software that they have used in the study. Also mention about the protein data bank.

Discussion part should be upgraded. 

Author Response

We gratefully appreciate all the comments and time of the reviewers and editor. We have considered all the comments in the manuscript, changes were marked in yellow for better location.

Details of the docking software that was used in the study, as well as the protein data bank was included in section 2.5 and in supplementary material. In discussion section, it was informed that TRA or PRA were explored on the molecular targets known not to be associated to their mechanism of action as it was confirmed by docking analysis

Reviewer 2 Report

This manuscript describes the pharmacological interaction of quercetin derivatives of Tilia americana for antihyperalgesic and antiallodynic effects. The study is well designed and executed. The experiments and presented data is in good flow. Few minor issues are there that needs to be addressed.

1. The introduction needs recent literature and references.

2. Conclusion is very brief. Add some more details.

Author Response

We are thankful for all your comments and suggestions. The answers are the following: 

1. The introduction needs recent literature and references.

ANSWER: Thank you for your comment. We have checked all the introduction text to include recent literature and references.

Reference 1 :

Former: Wood, P. Role of central dopamine in pain and analgesia. Expert. Rev. Neurother., 2008, 8, 781–797, doi:10.1586/14737175.8.5.781.

Current: Giorgi, V.; Sirotti, S.; Romano, M.E.; Marotto, D.; Ablin, J.N.; Salaffi, F.; Sarzi-Puttini, P. Fibromyalgia: One Year in Review 2022. Clin. Exp. Rheumatol. 2022, 40, 1065–1072, doi:10.55563/clinexprheumatol/if9gk2.

Reference 2:

Former:  Blasco-Serra, A.; Escrihuela-Vidal, F.; González-Soler, E.M.; Martínez-Expósito, F.; Blasco-Ausina, M.C.; Martínez-Bellver, S.; Cervera-Ferri, A.; Teruel-Martí, V.; Valverde-Navarro, A.A. Depressive-like symptoms in a reserpine-induced model of fibromyalgia in rats. Physiol. Behav., 2015, 151, 456-462, doi:10.1016/j.physbeh.2015.07.033.

Current: Jahan, F.; Nanji, K.; Qidwai, W.; Qasim, R. Fibromyalgia Syndrome: An overview of pathophysiology, diagnosis and management. Oman Med. J. 2012, 27, 192–195, doi:10.5001/omj.2012.44.

Reference 36

Former:  Russell, J.; Kamin, M.; Bennett, R.M.; Schnitzer, T.J.; Green, J.A.; Katz, W.A. Efficacy of tramadol in treatment of pain in fibromyalgia. J. Clin. Rheumatol., 2000, 6, 250–257, doi:10.1097/00124743-200010000-00004.

Current:  da Rocha, A.P.; Mizzaci, C.C.; Nunes Pinto, A.C.P.; da Silva Vieira, A.G.; Civile, V.; Trevisani, V.F.M. Tramadol for management of fibromyalgia pain and symptoms: Systematic review. Int. J. Clin. Pract. 2020, 74, e13455, doi:10.1111/IJCP.13455.

Reference 43:

Former:  Negri, G.; Santi, D.; Tabach, R. Flavonol glycosides found in hydroethanolic extracts from Tilia cordata, a species utilized as anxiolytics. Rev. Bras. Plantas Med., 2013, 15, 217–224, doi:10.1590/s1516-05722013000200008.

Current: Ullah, A.; Munir, S.; Badshah, S.L.; Khan, N.; Ghani, L.; Poulson, B.G.; Emwas, A.H.; Jaremko, M. Important flavonoids and their role as a therapeutic agent. Mol. 2020, 25, 1-39, doi:10.3390/MOLECULES25225243

2. Conclusion is very brief. Add some more details.

ANSWER: Thank you for your suggestion. Conclusion was reinforced in the text as follows:

In conclusion, central nervous system effects of Tilia americana var. mexicana are reinforced by the present results since significant antihyperalgesic and antiallodynic responses were demonstrated in an experimental FM model. The presence of quercetin derivatives flavonoids, as part of the responsible constituents, was confirmed into the brain tissue by epifluorescence corroborating their importance and potential bioactivity and supporting their influence as useful alternative of treatment for therapeutic of FM-type pain. However, their administration together with clinical analgesic drugs such as tramadol or pramipexol, both used for their moderate efficacy against this syndrome, is not recommended since they could reduce the benefits obtained in their individual administration.

Reviewer 3 Report

In order to explore the possible therapeutic alternative for FM-type pain, the analgesic effects of aqueous Tilia extract (TE) and its flavonoid fraction (FF) containing rutin and isoquercitrin were evaluated in this study by using RES-induced FM model in rats, alone and/or in combination with clinical drugs (TRA and PRA). TE(10-100 mg/kg, intraperitoneal injection) and FF(10-300 mg/kg, intraperitoneal injection) produced significant dose-dependent anti-hyperalgesic and antiallodynic effects, equivalent to TRA(3-10 mg/kg, intraperitoneal injection) or PRA(0.01-1 mg/kg, subcutaneous injection). However, according to the docking analysis, the combination of FF+TRA or FF+PRA leads to antagonistic interaction through the possible competition of serotonin transporter, μ or D2 receptor, respectively. This study provides a basis for the reasonable application of Tilia extract and is constructive. So, it is suggested to publish after minor revision.

1. It is necessary to briefly explain the meanings of subscripts in F4,33, F5,40, etc.

2. Why are there no docking results of TRA to D2 dopamine receptor and PRA to μ opioid receptor and serotonin transporter in molecular docking? A brief analysis should be needed even if the conclusion may be not supported.

3. Line 467. The number before “Threshold for the cold stimulus (cold allodynia)” should be 4.4.4.

Author Response

We are thankful for all your comments and suggestions. The answers are the following:  

1. It is necessary to briefly explain the meanings of subscripts in F4,33, F5,40, etc.

Thank you for your comment. The meaning of subscripts was included in the text as follows:

The F-value requires of two parameters, degrees of freedom of numerator (number of treatments minus one) and degrees of freedom denominator (number of experimental subjects minus number of treatments), where the intersection of numerator and denominator contains the critical F-value. It was included the F-value obtained in the corresponding one-way or two-way ANOVA analysis and the degrees of freedom (df) required to be contrasted with the F-table critical values for a significance level of 0.05.

2. Why are there no docking results of TRA to D2 dopamine receptor and PRA to μ opioid receptor and serotonin transporter in molecular docking? A brief analysis should be needed even if the conclusion may be not supported.

ANSWER: Thank you for your comment. Since D2 dopamine and μ opioid receptors and/or serotonin transporter are not molecular targets for TRA or PRA, respectively, we did not explore their interactions. However, we have included the docking analysis in supplementary material to demonstrate their low probability of interaction to the interested lectors.

3. Line 467. The number before “Threshold for the cold stimulus (cold allodynia)” should be 4.4.4.

ANSWER: Thank you for your observation. We have corrected the mistake in the sequence.